# Analysis of Spatial and Temporal Variability of Global Wetlands during the Last 20 Years Using GlobeLand30 Data

Mengjuan Li [1], Peng Ti [1,*], Xiuli Zhu [2], Tao Xiong [1], Yuting Mei [1] and Zhilin Li [1]

[1]  Faculty of Geosciences and Environmental Engineering, Institute of Sustainable Development of National Land, Southwest Jiaotong University, Chengdu 610032, China
[2]  National Geomatics Center of China, Beijing 100044, China
*   Correspondence: pti@swjtu.edu.cn

**Abstract:** Knowing the distributions and changes in global wetlands and their conversion to other land cover types could facilitate our understanding of wetland development, causes of variations, and decision-making for restoration and protection. This study aimed to comprehensively analyze the changes in wetland distributions at global, continental, typical regional, and national scales and the conversions between wetlands and other land cover types in the last 20 years. This study used GlobeLand30 (GL30) data with a 30 m resolution for the years 2000, 2010, and 2020. The main findings of this study are as follows: (1) the area of wetlands continued to increase globally from 2000 to 2020, with a total increase of approximately 4%. Wetland changes from 2010 to 2020 were more significant than those from 2000 to 2010. The regions with significant wetland changes were mainly in the north middle- and high-latitude, and the equatorial middle- and low-latitude, and Oceania and North America were the continents with the highest increase and decrease, respectively; (2) the major conversion of wetlands was mainly natural land cover types, including forest, grassland, water, and tundra, and there were minor conversions due to human activities, including the conversion of wetlands to cropland (~4600 km$^2$) and artificial land (~3400 km$^2$); (3) from 2000 to 2020, the increase in global wetlands was uneven, while the decrease was nearly even at a national scale. Australia had the highest increase due to the conversions from grass, bare land, and water, and Canada had the highest decrease due to the conversion into tundra and forest. The analysis results could more comprehensively characterize the distributions and changes of global wetlands, which may provide basic information and knowledge for related research work and policymaking.

**Keywords:** wetlands; change detection; land cover conversion; GlobeLand30; multiple spatial scales





## 1. Introduction

Wetlands are geographic complexes of aquatic biota and hydric soils that develop from wet or shallow flooded areas [1]. They are called the "kidneys of the Earth" as they play various roles in protecting global biodiversity [2], regulating global and local climate [3], surface water flow [4], etc. Wetlands offer fundamental materials for human well-being and have considerable economic value [5]. Changes in the area and distribution of wetlands may be directly or indirectly affected by natural and human factors, including climate change, sea-level rise [6], urbanization [7], farming [8], and regional development policies [9,10]. Knowing the distributions and changes in global wetlands, as well as the conversion to other land cover types, could facilitate our understanding of wetland development, causes of variations, and decision-making for restoration and protection [6–8]. Hence, the spatiotemporal characteristics of wetlands and their conversion to other land cover types have gained much attention from researchers and policymakers [3,11].

Matthews and Fung [10] developed a global wetland database at 1° resolution and analyzed the global distribution of different wetland groups. The World Wildlife Fund and Center for Environmental System Research, University of Kassel, Germany, developed the

Global Lakes and Wetlands Database (GLWD) with a resolution of ≥1 km, specifically for wetlands [12]. However, these databases contain only wetland information and cannot support the conversion analysis between wetlands and other land types. Much research on wetland change assessments has been conducted using remote sensing images that contain information on other land cover types, while mainly focusing on some local regions, such as the Great Lakes [13], Lake Chad [14], and Dong ting Lake [15]. Some researchers have used data compilation from national wetland inventories to achieve global wetlands assessment and inventory [16,17]. For instance, according to published data, Davidson [18] summarized the changes in wetland distribution at different spatial scales, such as global, continental, and national scales. Nevertheless, in these compilation work studies, there were still some limitations of adequate inventory for some countries and uncertainties in the analysis results owing to the inconsistency in the basic data collected from different countries and studies [5,11].

Analysis of global and regional changes in the world's remaining wetlands is challenging. The lack of a global high-resolution wetland dataset and the conversion information between wetland and other land cover types results in the uncertainty of analysis results and limitations to the analysis of wetland changes. GlobeLand30 (GL30), the first global land cover dataset at a 30 m resolution [19], provides an alternative for such an analysis. Furthermore, the GlobeLand30 data contain ten land cover types, including wetland, forest, grassland, shrub, cropland, water, tundra, artificial land, bare land, and ice [20], and can support the conversion analysis among different land cover types. Many researchers have also used GL30 data to analyze the distributions and changes in different land cover types, such as cropland and water [21–23]. Based on GlobeLand30 data for 2000, 2010, and 2020, this study aims to comprehensively analyze the changes in wetland distribution on different spatial scales, including global, continental, typical regional, and national spatial scales, and the conversions between wetlands and other land cover types in the last 20 years. This study used three assessment metrics, namely, the area change, change rate, and importance index of change, and evaluation methods, including the conversion matrix of land types and the Lorenz curve. Compared with those of existing research, the results of this study could more comprehensively characterize the distributions and changes in global wetlands [20,24,25], allowing us to better understand how wetlands have changed globally over the last 20 years and providing basic information and knowledge for a variety of analysis purposes, such as biodiversity, greenhouse gas emissions from wetlands, local and global climate, and wetland management.

## 2. Data and Methods

### 2.1. Data

GlobeLand30 is the world's first 30 m resolution global land cover data product developed by the National Geomatics Center of China, with the TM5, ETM+, and OLI multispectral images of the US Landsat, China Environmental Disaster Mitigation Satellite HJ-1 Multispectral image, and GaoFen-1 (GF-1) multispectral image [19,20], which contains information on global land cover from 80°N to 80°S [20]. The evaluation results showed that the overall accuracies of the 2000, 2010, and 2020 data were 83.50%, 85.72%, and 85.72%, respectively [20,26]. Wetlands in GlobeLand30 are defined as land covered with wetland plants and water, including inland marshes, lake marshes, river floodplain wetlands, forest/shrub wetlands, peat bogs, mangroves, and salt marshes. In some other definitions of wetland type in related work, more subtypes of land cover are contained. For instance, in the Ramsar Convention, the wetland type contains some permanent rivers and lakes, which were defined as water bodies in GlobeLand30 [27]; tundra and seasonally flooded agricultural land are often regarded as wetland type [27–29].

The vector data of continent boundaries used in this study were obtained from the National Earth System Science Data Center, National Science & Technology Infrastructure of China (http://www.geodata.cn (accessed on 12, November, 2021)). To extract the change information from GlobeLand30 data, data mosaicking was conducted using ArcGIS

software and a Python program was developed to calculate the statistical metrics described in Section 2.2 at different spatial scales.

### 2.2. Characteristic Metrics of Wetland Changes

The evaluation metrics and methods used in this study to analyze the spatial and temporal changes in global wetlands included (1) three evaluation metrics: change in area, change rate, and importance index of change; (2) Lorenz curves and Gini coefficients were used to estimate the equality of global wetland changes; (3) the land cover type conversion matrix was used to quantitatively describe the conversion between wetland and other land cover types. The metrics and methods are described as follows.

#### 2.2.1. Change Assessment Metrics

(1) Change in Area

The change in the area metric was used to calculate the net change in the area of the target land cover type in a statistical unit during a certain period. This was calculated as follows:

$$S_{a_i - b_i} = S_{a_i} - S_{b_i} \tag{1}$$

where $S_{a_i - b_i}$ represents the change in the area of land cover type $i$ from year $a$ to year $b$ and $S_{a_i}$ and $S_{b_i}$ are the areas of land cover type $i$ in years $a$ and $b$, respectively.

(2) Rate of Change

The rate of change metric reflects the extent of change in the target land cover type during a certain period and was calculated as follows:

$$\mathrm{K} = \frac{S_{ai - bi}}{S_{ai}} \times 100\% \tag{2}$$

where K is the rate of change from year $a$ to year $b$; $S_{a_i - b_i}$ represents the change in the area of land cover type $i$ from year $a$ to year $b$; $S_{a_i}$ is the area of land cover type $i$ in year $a$.

(3) Change Importance Index

The change importance index can be used to represent the importance of different types of land cover in the conversion, and a larger $index_i$ shows that the conversion between the target land cover type and the $i$th land cover type is more dominant [30]. This was calculated as follows:

$$\begin{aligned} index_i &= \frac{A_i}{A} \times 100\% \\ A\& &= \sum_{i=1}^{n} A_i \end{aligned} \tag{3}$$

where $index_i$ is the importance index of the conversion between the target land cover type and the $i$th land cover type, varying from 0% to 100%; $A_i$ is the area of the target land cover converted to and from the $i$th land cover; $A$ is the sum of the areas of the target land cover converted to and from other land cover types.

#### 2.2.2. Lorenz Curve and Gini Coefficient

The Lorenz curve was first used to compare and analyze the equality of income distribution [31,32]. It has been widely used to analyze the spatial distributions of land cover and the equality of spatial changes [33–35]. This study used the Lorenz curve to assess the equality of wetland changes at a global scale. To generate the Lorenz curve, the change rates of countries were sorted in ascending order, and then the cumulative percentages of the original and change areas were calculated and represented by the x- and y-axes. The Lorenz curve formed is as illustrated in Figure 1; the closer the curve is to y = x, the smaller the curvature, indicating that the change is more equal.

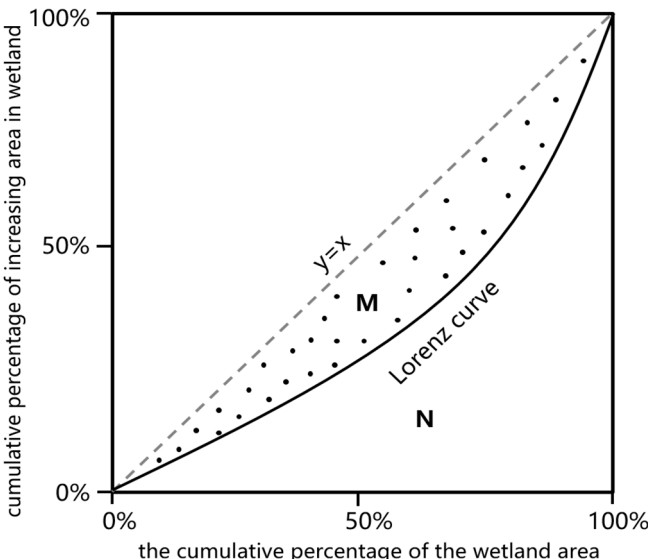

**Figure 1.** The Lorenz curve.

The Gini coefficient, also called the Lorenz coefficient, was used to quantitatively assess the equality of the Lorenz curve. As shown in Figure 1, the Gini coefficient is the ratio of the total area below the line of equality (the sum of the areas of M and N) divided by the area between the line of y = x and the Lorenz curve (area M) [32]. The Gini coefficient ranges between 0 and 1, and a value of 0 indicates complete equilibrium [32,33]. The Gini coefficient below 0.2 indicates a perfect equality, and 0.2–0.3 indicates a relative equality, 0.3–0.4 indicates a slight equality, 0.4–0.5 indicates a relative inequality, and >0.5 indicates a significant inequality [36]. The Gini coefficient was calculated as follows:

$$G = 1 - \sum_{i=1}^{n}(x_{i+1} - x_i)(y_{i+1} + y_i) \tag{4}$$

where G is the Gini coefficient; $x_i$ and $y_i$ are the cumulative percentages of the original areas and the change areas, respectively; $n$ is the total number of statistical units where changes occurred.

### 2.2.3. Land Cover Conversion Matrix

The land cover conversion matrix contains information on the conversion between the target land type and other land cover types [37]. In this study, the area $S_{iw}$ of conversion between wetlands and other land types was used as a variable to describe the direction and evolution of the conversion of wetlands and other land cover types within a certain spatial and temporal range in the form of a matrix.

$$S_{i-w} = \&\begin{bmatrix} S_{1w} & S_{w1} \\ \vdots & \vdots \\ S_{nw} & S_{wn} \end{bmatrix} \tag{5}$$
*Converted to wetland Converted from wetland*

where $S_{i-w}$ is the conversion area between land cover type i and wetland within the study period and region; $w$ represents wetland; $i$ represents other land cover type; $S_{iw}$ represents the conversion area from land cover type $i$ to wetland; $S_{wi}$ represents the conversion area from wetlands to land cover type $i$.

## 3. Results and Analysis

### 3.1. Changes in Global Wetlands

In this subsection, an analysis of the distributions and changes in wetlands from 2000 to 2020 at global, continental, and regional-with-significant-changes scales is presented. According to the statistical results obtained using GL30 data, the total global wetland area was approximately 3.44 million km$^2$ in the year 2000. As shown in Figure 2, most wetlands were distributed in the middle-high latitudes of the northern hemisphere and the middle-low latitudes of the southern hemisphere. Some typical regions with larger areas of wetlands (depicted by red elliptical circles in Figure 2) included: (1) the Amazon River Basin region, and in this region, there is the world's largest wetland, that is, the Pantanal [38]; (2) the West Siberian lowlands of Russia; (3) the Canadian provinces of Manitoba and Ontario near Hudson Bay; (4) the region of America near Florida. Some distribution characteristics were reported by related analysis work. For instance, Zedler and Kercher [3] found that Florida, the West Siberian lowlands, the Amazon River, and the Hudson Bay lowlands have larger wetland areas; Hu et al. [11] found similar wetland spatial distribution characteristics in the latitude direction; Mitsch and Hernandez [28] indicated that the greatest Canadian wetlands were in the provinces of Manitoba and Ontario.

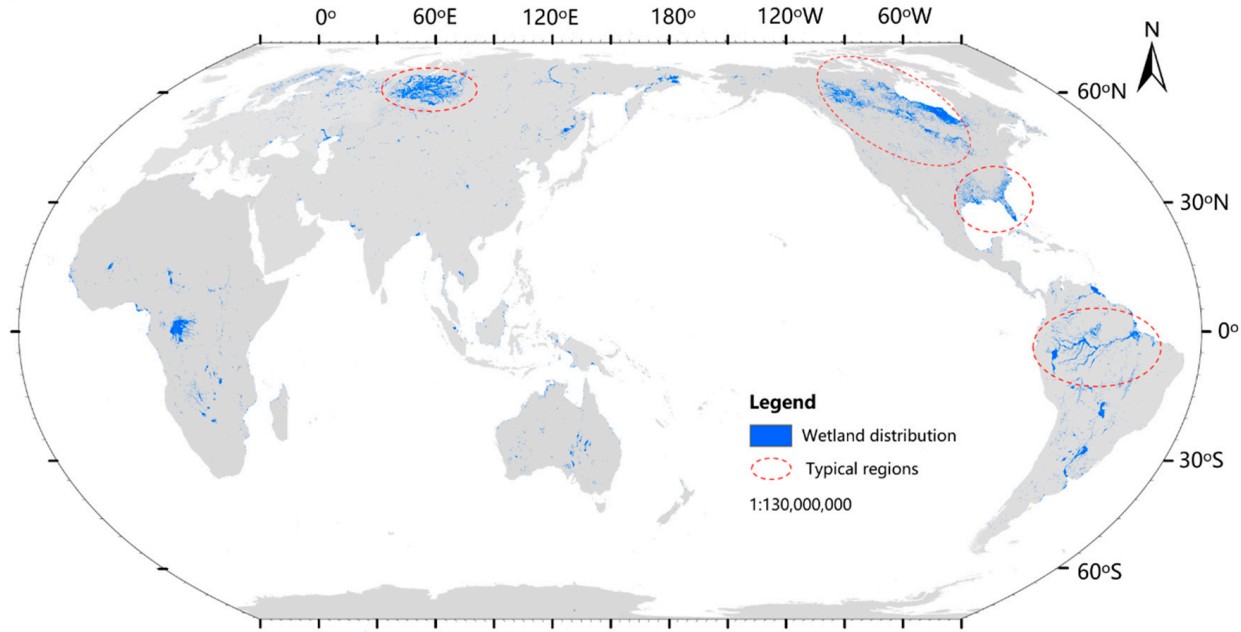

**Figure 2.** The spatial distribution of global wetlands in 2000.

According to Table 1, in 2000, North America has the largest wetland area of ~1.09 million km$^2$, followed by Asia with ~0.87 million km$^2$ and South America of ~0.72 million km$^2$. Oceania has the smallest wetland area of ~0.09 million km$^2$. Some related work indicated that Asia has the largest wetland area, while there was a large area of seasonally inundated rice paddies in this region [18,39], which may be classified as cropland in GL30 data.

**Table 1.** Changes in wetland areas by continent from 2000 to 2020.

|  | Oceania | Africa | Asia | South America | Europe | North America | Total |
|---|---|---|---|---|---|---|---|
| Area in 2000 (10$^4$ km$^2$) | 9.94 | 39.96 | 87.15 | 72.07 | 18.51 | 109.26 | 344.06 |
| 2000 to 2010 (10$^4$ km$^2$) | +0.86 | −1.24 | +1.23 | +1.75 | +0.74 | −0.94 | +2.4 |
| 2010 to 2020 (10$^4$ km$^2$) | +7.78 | +9.18 | +4.06 | +1.65 | −0.97 | −11.46 | +10.24 |
| 2000 to 2020 (10$^4$ km$^2$) | +8.64 | +7.94 | +5.29 | +3.4 | −0.23 | −12.40 | +12.64 |

From 2000 to 2020, global wetlands experienced a total net increase of approximately 126,400 km$^2$. The continents with a net increase in wetland areas were Oceania, Africa, Asia, and South America. Among them, Oceania had the highest increase rate (87%) and a net increase in wetland area (86,400 km$^2$). North America had the highest rate of decrease (11%) and net area decrease (−124,000 km$^2$). As the area change of global wetlands from 2010 to 2020 (102,400 km$^2$) was much greater than that from 2000 to 2010 (+24,000 km$^2$), the changes occurred mainly in the period from 2010 to 2020. Some research work for some local regions also found that the change in wetlands was not significant from 2000 to 2010, for example, the Harike wetland in India [40] and wetlands in China [41]. As illustrated in Figure 3, from 2000 to 2020, there were several regions with significant changes in wetlands and their locations are depicted by red labels in Figure 3a and red elliptical circles in Figure 3b: (1) the northern Eurasian; in this region, the wetland changes in central and northern West Siberia were considerably significant [39]; (2) the region of Canada near the Hudson Bay; (3) the region near the Gulf Coast of southern United States; (4) the central plains of Oceania.

### 3.2. Analysis of the Conversions between Wetland with Other Land Cover Types

This subsection focuses on the characteristic analysis of the interconversions between wetlands and other land-cover types from 2000 to 2020 at global, continental, typical regional, and national scales.

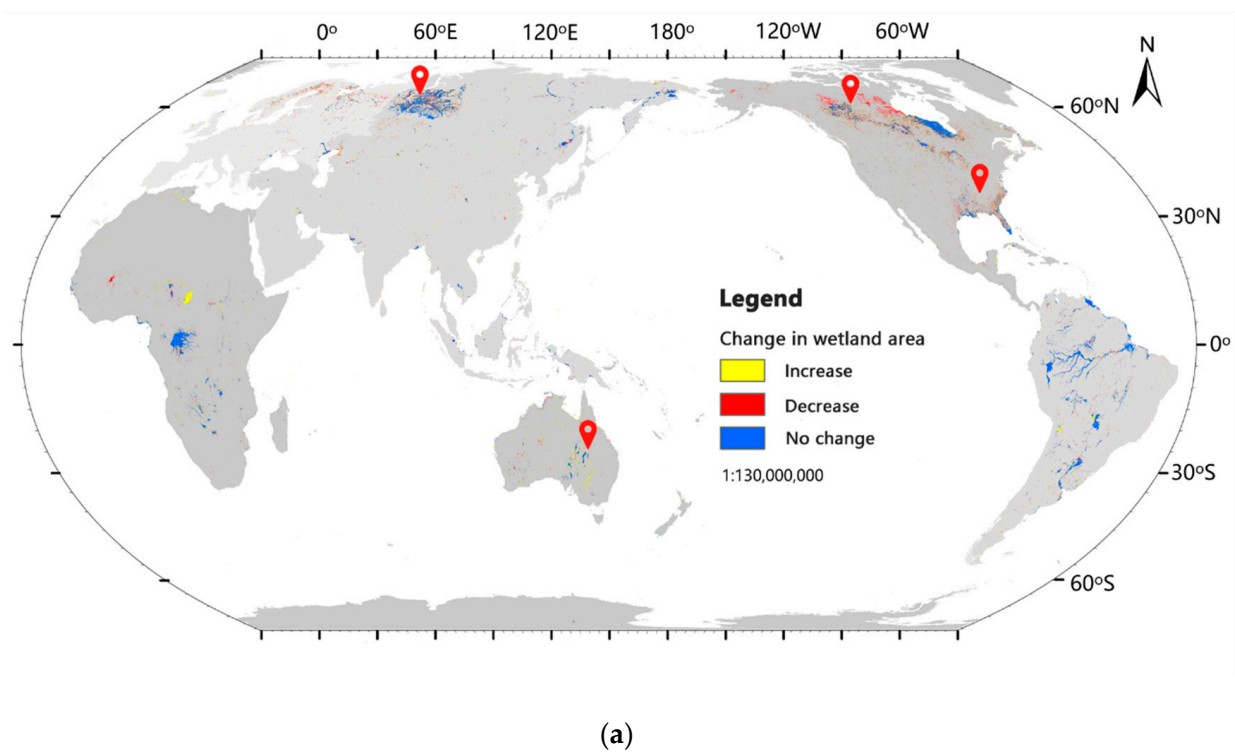

(**a**)

**Figure 3.** *Cont.*

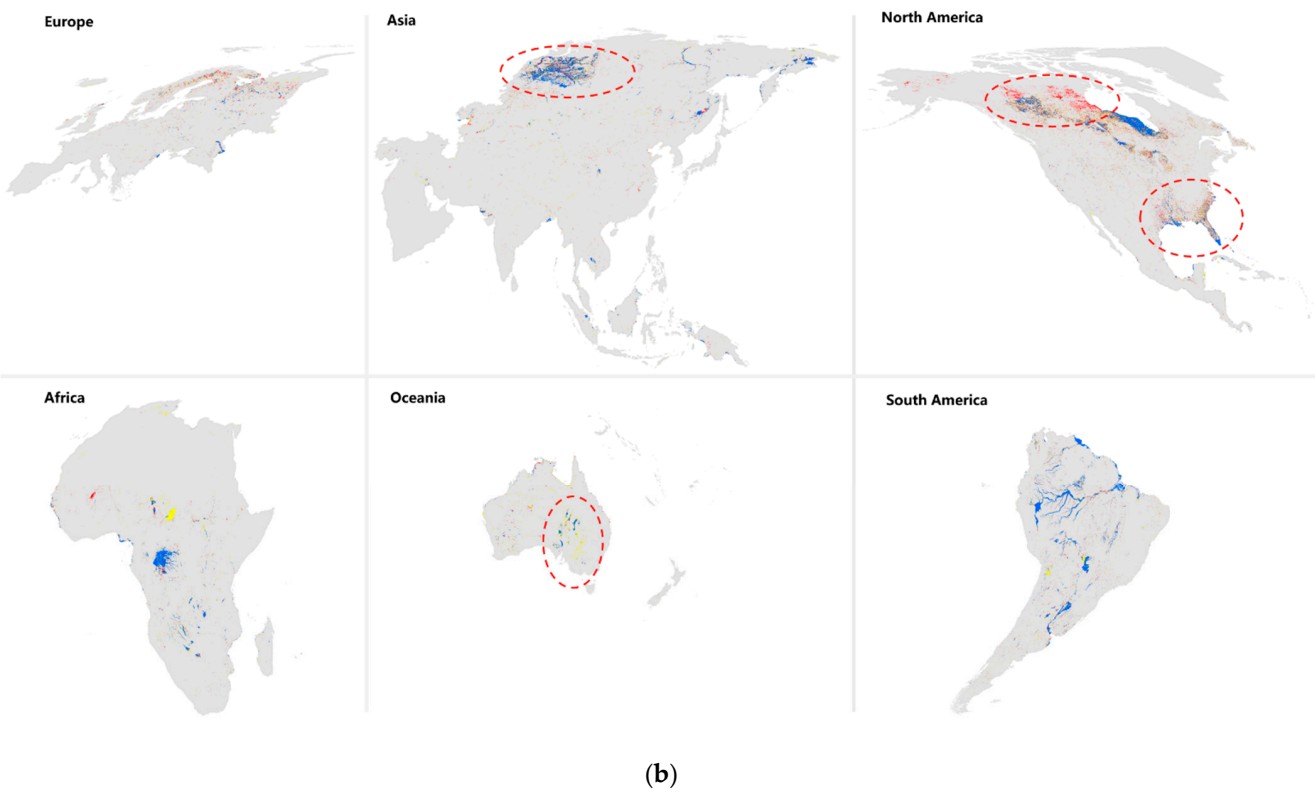

(**b**)

**Figure 3.** Changes in global wetlands from 2000 to 2020. (**a**). Changes in global spatial distribution of wetlands from 2000 to 2020. (**b**). Changes in global spatial distribution of wetlands from 2000 to 2020, by continent.

From 2000 to 2020, the total area of wetlands converted from other land types was approximately 771,700 km$^2$, and that converted to other land types was approximately 645,300 km$^2$. According to Figure 4 and Table 2, the conversion of global wetlands from 2000 to 2020 can be described as follows:

- Four land cover types, including forest, grass, water, and tundra, were prominent in the conversion to wetlands, accounting for more than 70% and 80% of the increase and decrease areas of wetlands, respectively, according to the conversion importance indices shown in Table 2.
- The conversion from forest, grass, and bare land led to a significant net increase in wetlands, and the conversion from wetland to tundra led to a significant net decrease in wetlands. The conversions between wetlands and other land-cover types were almost balanced.
- According to the major types converted with wetlands, it may indicate that the conversions may be mainly affected by natural factors, such as wetland degradation due to climate warming resulting in conversion to tundra, grass, etc. [28,42]; water expansion submerging the surrounding grass and forest, resulting in the conversion to wetlands [43]; sea-level rise resulting in the loss of coastal wetlands [42,44,45].

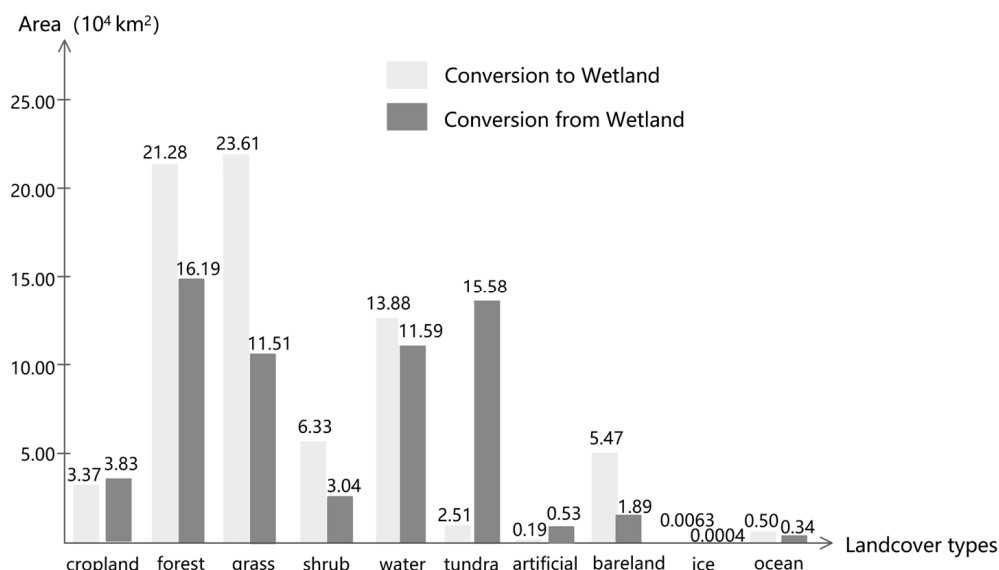

**Figure 4.** Global conversions between wetland and other land cover types, 2000–2020.

**Table 2.** Conversions between wetlands and other land cover types from 2000 to 2020.

| Landcover types | Cropland | Forest | Grass | Shrub | Water | Tundra | Artificial | Bare land | Ice | Ocean | Total |
|---|---|---|---|---|---|---|---|---|---|---|---|
| Conversion to Wetland ($10^4$ km²)/ Conversion important index | 3.37 / 4.37% | 21.28 / 27.58% | 23.61 / 30.59% | 6.33 / 8.20% | 13.88 / 17.99% | 2.51 / 3.25% | 0.19 / 0.25% | 5.47 / 7.09% | 0.01 / 0.01% | 0.50 / 0.65% | 77.17 / 100% |
| Conversion from Wetland ($10^4$ km²)/ Conversion important index | 3.83 / 5.94% | 16.19 / 25.09% | 11.51 / 17.84% | 3.04 / 4.72% | 11.59 / 17.96% | 15.58 / 24.14% | 0.53 / 0.83% | 1.89 / 2.93% | ≈0.00 / 0.00% | 0.34 / 0.53% | 64.53 / 100% |

The maps in Figure 5a,b show the spatial distribution of the increase and decrease in wetland area of the conversions with other land types from 2000 to 2020, respectively. As illustrated in Figure 5a,b, the wetland conversions with other land cover types had an uneven spatial distribution. Specifically, forest-to-wetland and grassland-to-wetland conversions were mainly in Asia (30°N–65°N, 60°E–90°E) and South America (20°S–10°N, 30°W–80°W) in the tropical climate zone, resulting in an increase in global wetland areas. The wetland-tundra conversions were mainly in the continental subarctic climate zone near Hudson Bay, Canada in North America (55°N–65°N, 60°W–120°W), which resulted in a considerable decrease in North America. The conversion between wetlands and water was almost balanced, but the spatial distribution of the conversion was not homogenous, and the wetland-to-water converted area was more than that of the water-to-wetland in North America and the opposite of that in Oceania. Detailed information on the prominent land cover types in the conversion to wetlands is as follows:

- Figure 5a shows the increase in wetland area and the details of other land cover types converted to wetlands from 2000 to 2020, including: (1) the forest-to-wetland conversions were mainly in North America, Asia, and South America, with a total conversion area of 163,400 km², accounting for approximately 77% of the total global forest-to-wetland conversion area; (2) the grassland-to-wetland conversions were mainly in Africa, Asia, and Oceania, with a total area of 186,900 km², accounting for approximately 79% of the total global grassland-to-wetland conversion area; (3) the water-to-wetland conversions were mainly in South America, Asia, and Oceania, with a total conversion area of 103,200 km², accounting for approximately 74% of the total global water-to-wetland conversion area. (4) The shrub-to-wetland conver-

sions occurred mainly in North America, South America, and Africa, with a total area of 53,200 km$^2$, accounting for approximately 84% of the total global shrub-to-wetland conversion area.

- Figure 5b shows the decrease in wetland area and the details of wetlands converted to other land cover types from 2000 to 2020, including: (1) The wetland-to-forest conversions were mainly concentrated in North America, South America, and Asia, with a total conversion area of 127,500 km$^2$, accounting for about 79% of the total global wetland-to-forest conversion area. (2) The wetland-to-grass conversions were mainly in Asia, Africa, and South America, with a total conversion area of 83,800 km$^2$, accounting for about 73% of the total global wetland-to-grassland conversion area. (3) The wetland-to-water conversions were mainly distributed in Asia, North America, and South America, with a total conversion area of 96,100 km$^2$, accounting for about 83% of the total global wetland-to-water body conversion area. (4) The wetland-to-tundra conversions occurred only in North America, Europe, and Asia, with a total area of 155,800 km$^2$. (5) The wetland-to-shrub conversions were mainly in North America, South America, and Africa, with a total area of 26,500 km$^2$, accounting for approximately 87% of the total wetland-to-shrubland conversion area.

For more details, an analysis of wetland conversion for a short time interval from 2000 to 2010 and 2010 to 2020 was conducted. According to Figures 6 and 7 and Table 3, the conversion of global wetlands can be described as follows:

- From 2000 to 2010 and 2010 to 2020, the total area of wetlands converted from other land types was approximately 366,200 km$^2$ and 405,500 km$^2$, respectively, and the total area of wetlands converted to other land types was 342,200 km$^2$ and 303,100 km$^2$, respectively.
- Four land cover types, including forest, grass, water, and tundra, were prominent in the conversion to wetlands for both periods, according to the conversion importance indices shown in Table 3.
- According to Figure 6, from 2000 to 2010, the conversion from water and tundra led to a net increase in wetlands, the conversion of wetland-to-grass led to a net decrease in wetlands, and the conversions between wetland and other land cover types were almost balanced. According to Figure 7, from 2010 to 2020, the conversion from forest, grass, shrub, and bare land led to a net increase in wetlands; the conversion of wetland-to-tundra led to a net decrease in wetlands; conversions between wetland and other land cover types were almost balanced.
- By comparing Figure 6 with Figure 7, it can be found that: (1) the conversion areas of grass-to-wetland from 2010 to 2020 contributed considerably to the net increase in wetlands, which was more than six times that of the wetland-to-grass, while there was little difference from 2000 to 2010, and forest and shrub had almost the same situation; (2) the conversion areas of wetland-to-tundra were significantly more than those of tundra-to-wetland from 2010 to 2020, which had a considerable contribution to the net decrease in wetlands, while the opposite was true from 2000 to 2010.

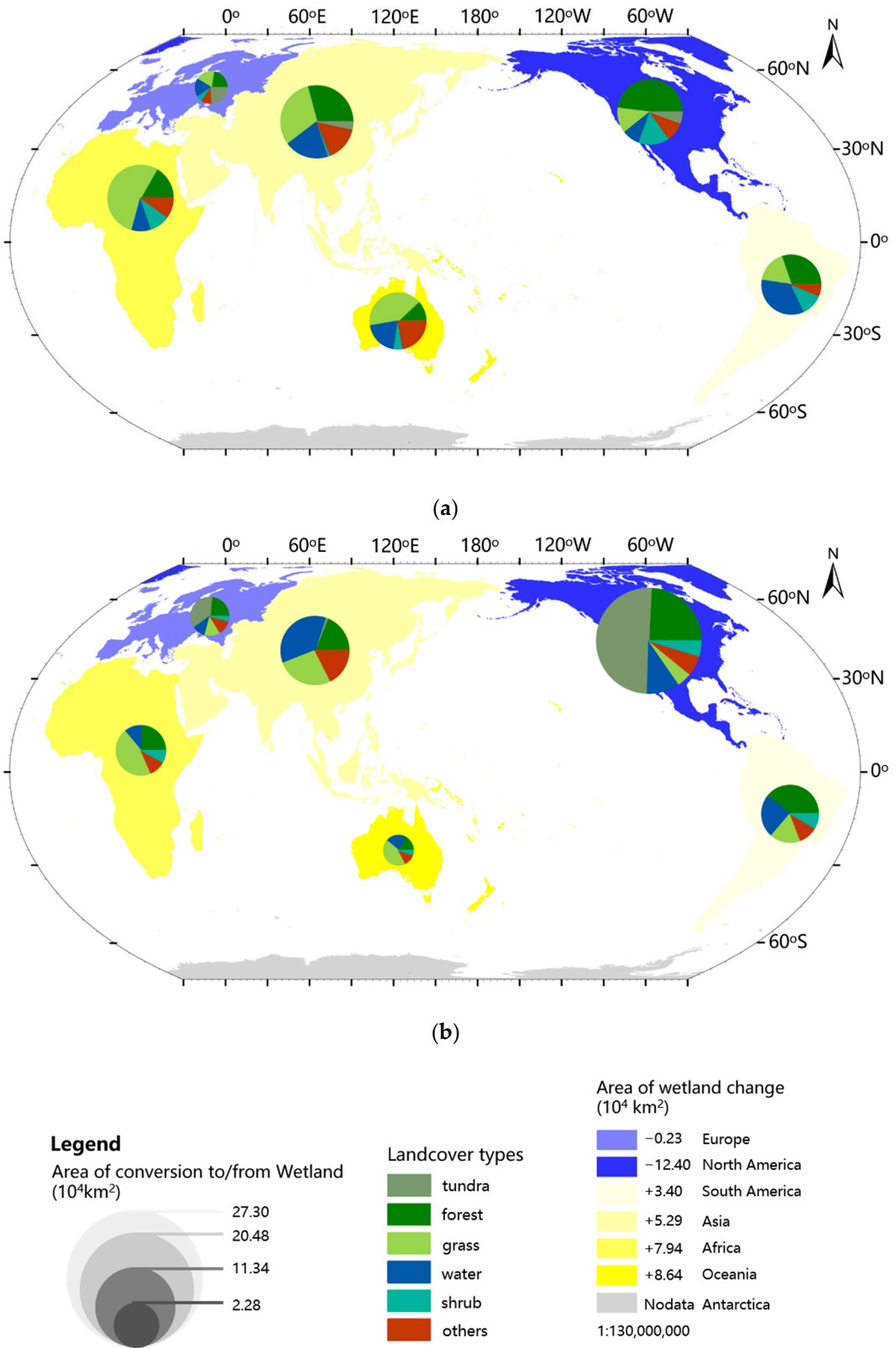

**Figure 5.** Wetlands conversions with other land cover types by continent, 2000–2020. (**a**). Wetlands converted from other land cover types by continent, 2000–2020. (**b**). Wetlands converted to other land cover types by continent, 2000–2020.

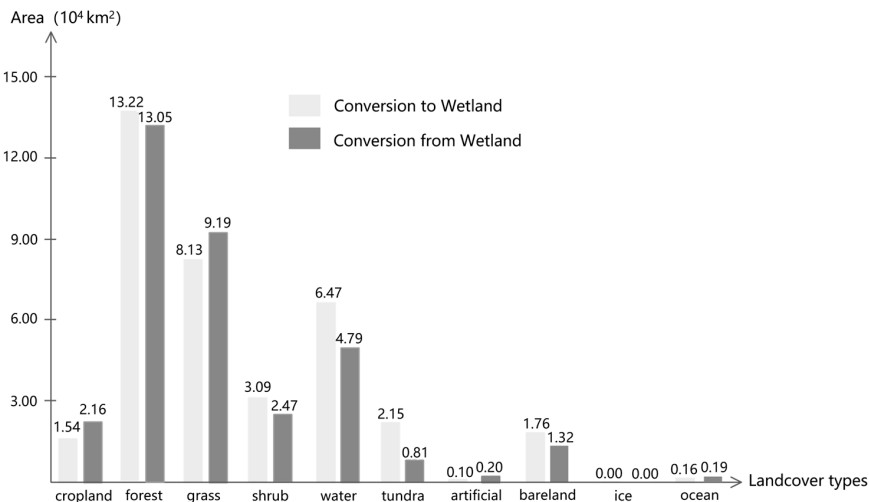

**Figure 6.** Wetlands conversions with other land cover types by continent, 2000–2010.

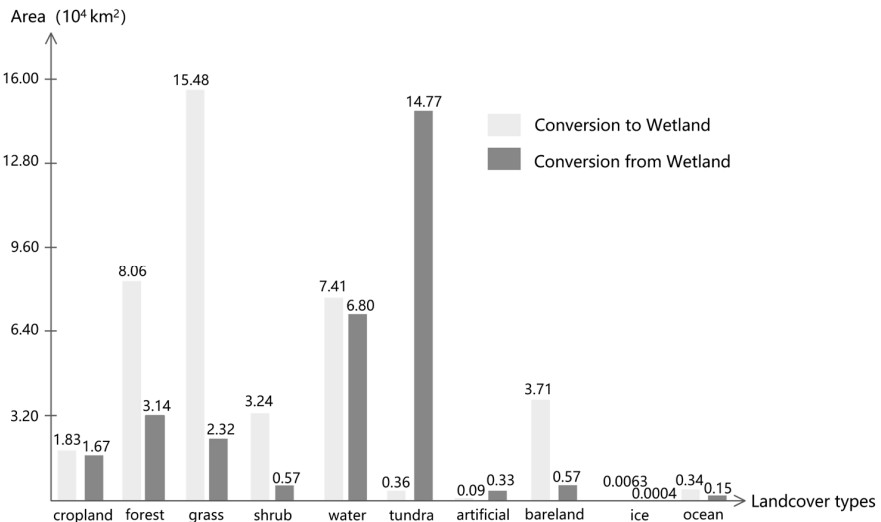

**Figure 7.** Wetlands conversions with other land cover types by continent, 2010–2020.

**Table 3.** Conversions between wetlands and other landcover types in two periods from 2000 to 2020.

| Landcover Types | 2000–2010 | | 2010–2020 | |
|---|---|---|---|---|
| | Conversion to Wetland (10⁴ km²)/Conversion Important Index | Conversion from Wetland (10⁴ km²)/Conversion Important Index | Conversion to Wetland (10⁴ km²)/Conversion Important Index | Conversion from Wetland (10⁴ km²)/ Conversion Important Index |
| Cropland | 1.54/4.21% | 2.16/6.31% | 1.83/4.51% | 1.67/5.51% |
| Forest | 13.22/36.10% | 13.05/38.14% | 8.06/19.88% | 3.14/10.36% |
| Grass | 8.13/22.20% | 9.19/26.86% | 15.48/38.18% | 2.32/7.65% |
| Shrub | 3.09/8.44% | 2.47/7.22% | 3.24/7.99% | 0.57/1.90% |
| Water | 6.47/17.67% | 4.79/14.00% | 7.41/18.27% | 6.80/22.43% |
| Tundra | 2.15/5.87% | 0.81/2.37% | 0.36/0.89% | 14.77/48.73% |
| Artificial | 0.10/0.27% | 0.20/0.58% | 0.09/0.23% | 0.33/1.10% |
| Bareland | 1.76/4.81% | 1.32/3.86% | 3.71/9.15% | 0.57/1.88% |
| Ice | 0.00/0.00% | 0.00/0.00% | 0.01/0.02% | 0.0004/0.0014% |
| Ocean | 0.16/0.44% | 0.19/0.56% | 0.34/0.84% | 0.15/1.51% |
| Total | 36.62 (10⁴ km²) | 34.22 (10⁴ km²) | 40.55 (10⁴ km²) | 30.31 (10⁴ km²) |

The maps in Figures 8 and 9 show the increase and decrease in wetland area in different continents and the conversions with other land types from 2000 to 2010 and 2010 to 2020, respectively.

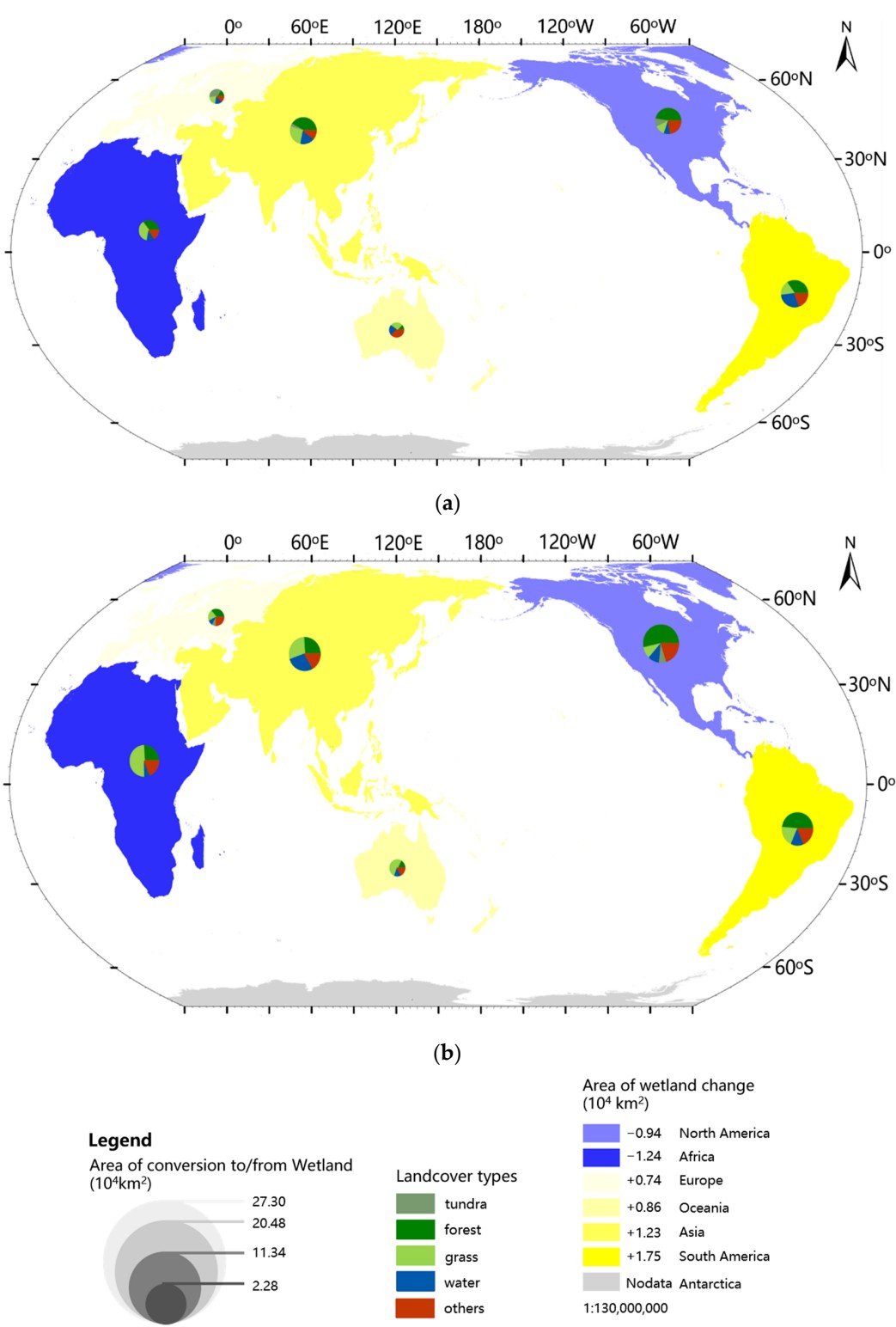

**Figure 8.** Wetlands conversions with other land cover types by continent, 2000–2010. (**a**). Wetlands converted from other land cover types by continent, 2000–2010. (**b**). Wetlands converted to other land cover types by continent, 2000–2010.

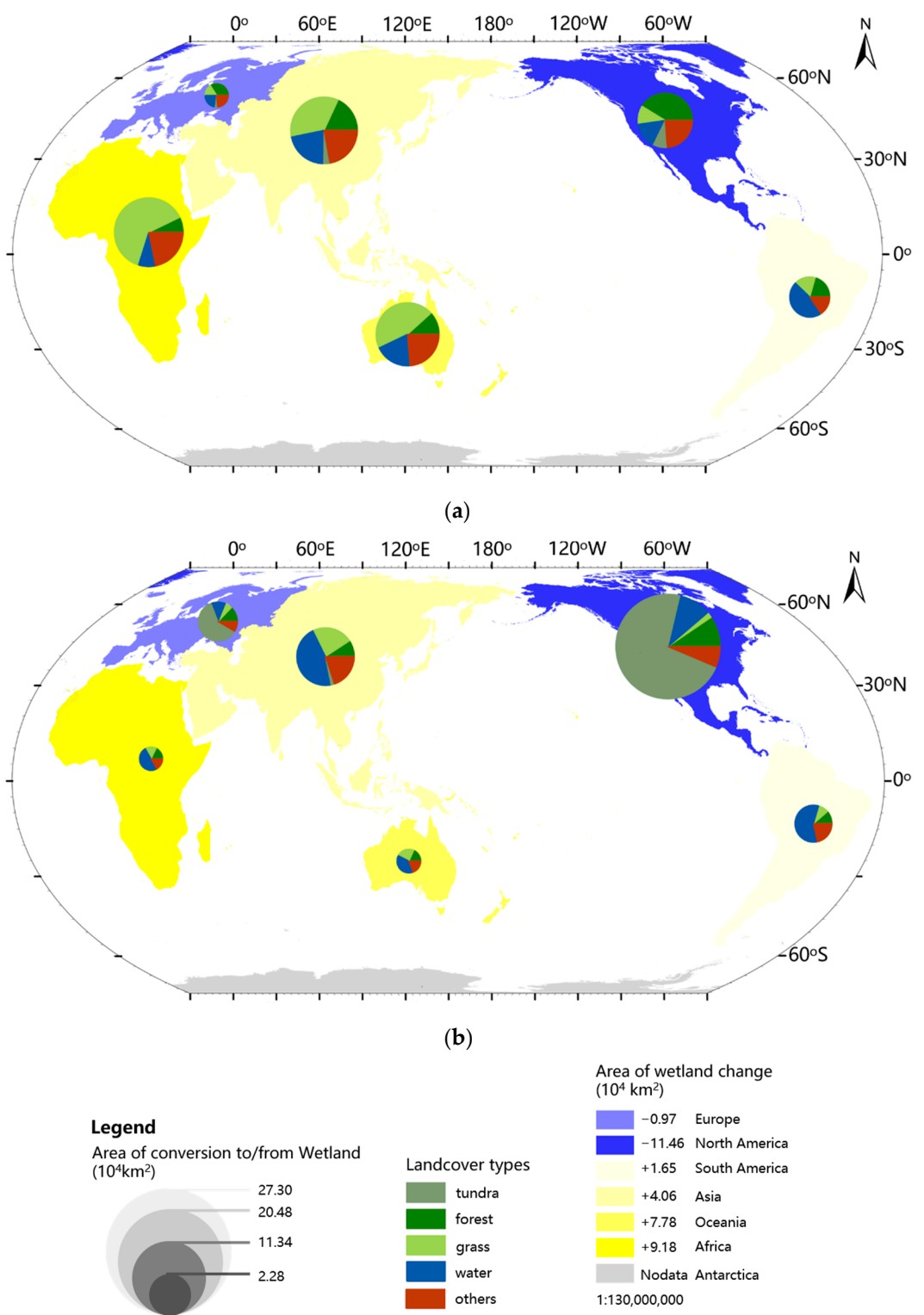

**Figure 9.** Wetlands conversions with other land cover types by continent, 2010–2020. (**a**). Wetlands converted from other land cover types by continent, 2010–2020. (**b**). Wetlands converted to other land cover types by continent, 2010–2020.

- By comparing Figures 8 and 9, it can be seen that the wetland conversion in each continent from 2000 to 2010 was significantly smaller than that from 2010 to 2020.
- As illustrated in Figure 8, North America and Africa had net decreases in wetland areas, while other continents had net increases in wetland areas from 2000 to 2010. By comparing Figure 8a with Figure 8b, it can be found that the main types of land covers

converted with wetland and the amounts of the conversions to/from wetland were almost the same in each continent.

- As illustrated in Figure 9, North America and Europe had net decreases in wetland areas, whereas other continents had net increases in wetland areas from 2010 to 2020. By comparing Figure 9a with Figure 9b, it can be found that except South America, there were larger differences for the main types of land covers converted with wetland and the amounts of the conversions to/from wetland in other continents during the period.
- As illustrated in Figure 9b, the conversion areas of wetland-to-tundra in North America and Europe were much larger, which may be due to the melting of permafrost according to the analysis work of Mitsch and Hernandez (2013) [28].

### 3.3. Analysis on Equality of Wetland Changes of Global Countries

This subsection analyzes the spatial equality of area changes at the national scale using the Lorenz curve and the conversion situations between wetlands with other land cover types.

Figure 10a,b show two Lorenz curves for increases and decreases in the wetland area from 2000 to 2020, respectively. In Figure 10a, the Lorenz curve has a high curvature, and the Gini coefficient is 0.73 (>0.5 indicates a significant inequality). This indicates that the increase in global wetland area was very uneven; approximately 10% of the countries had more than 70% of the increase in global wetland area. Comparatively, the Lorenz curve of the decreases in wetland area in Figure 10b has a small curvature and the Gini coefficient is 0.29 (0.2–0.3 indicates a relative equality), which indicates that the decrease in global wetland area was nearly even.

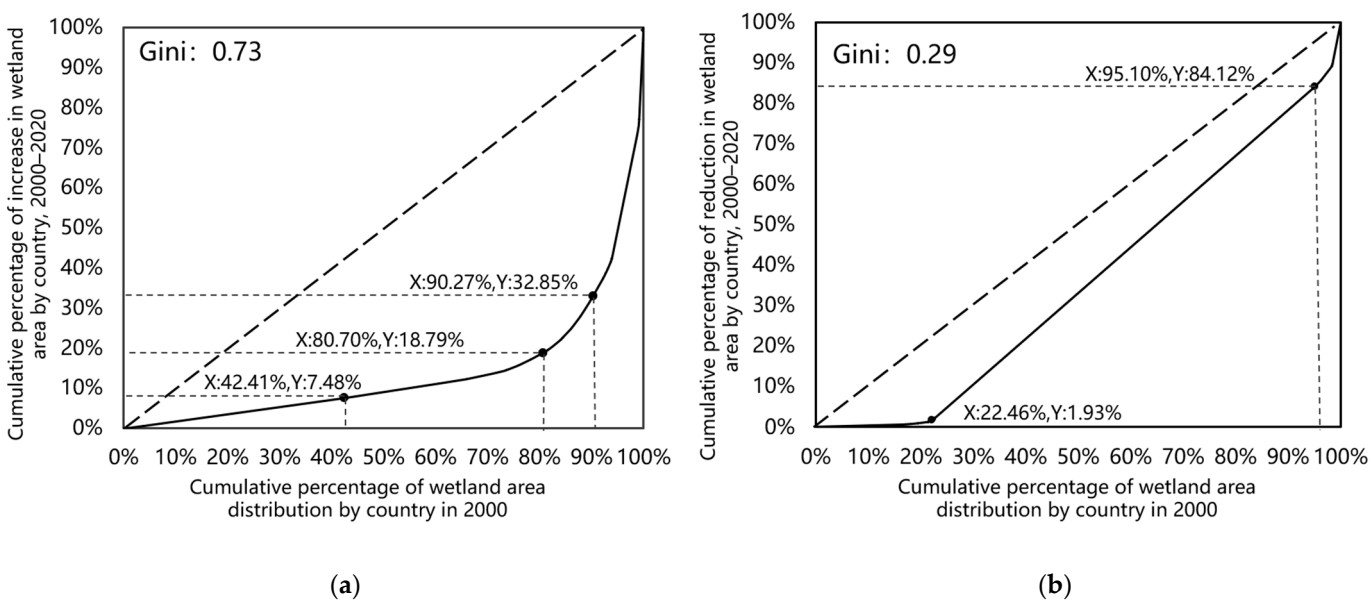

**Figure 10.** Lorenz curves representing the equality of wetland area changes from 2000 to 2020. (**a**). Lorenz curve for the area increase. (**b**). Lorenz curve for the area decrease.

As shown in Figure 11, the area decreases mainly occurred in the countries in North America, some countries in Europe and Africa, and several countries in Asia. The three countries with the highest decreases in the global wetland area were Canada, Mali, and Finland, and they had 96% of the total decrease in global wetland area. The decrease in wetlands in Canada has also been reported by Mitsch and Hernandez (2013) [28]. Approximately 60% of the world's countries had an increase in wetland areas, and the three countries with the highest increases in wetland area were Australia, Chad, and Russia, with 56% of the increase in global wetland area. According to related studies, Chad and Russia

have a large area of wetlands under the Ramsar Convention's protection [46], and the area of wetlands in these two countries has increased [47,48].

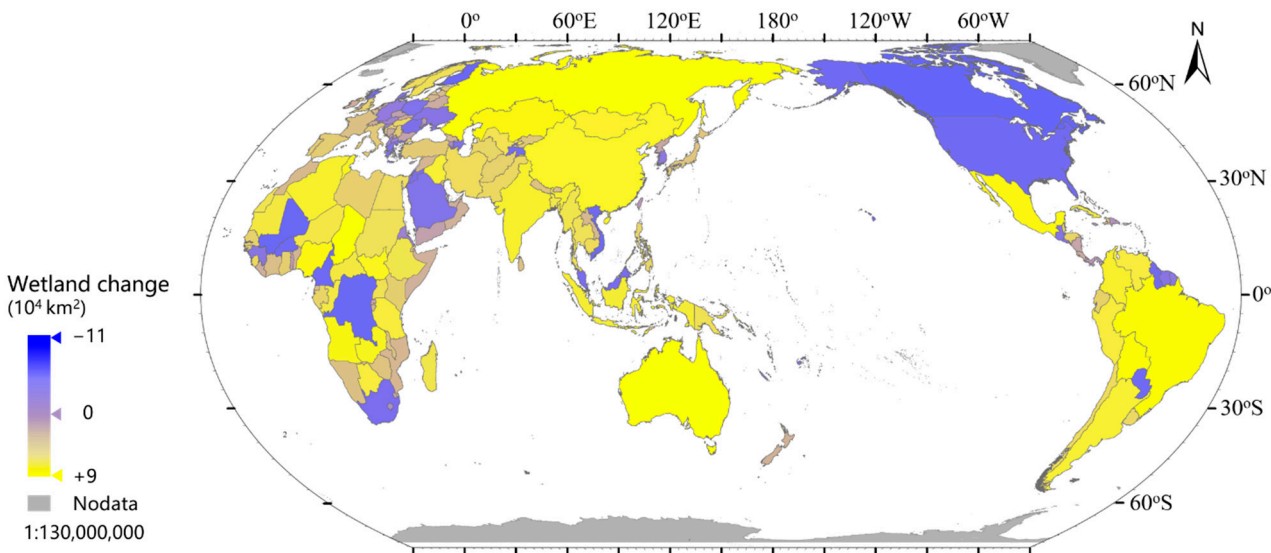

**Figure 11.** The wetland area changes of global countries from 2000 to 2020.

From 2000 to 2020, the three countries with the largest increases in wetland area were Australia, Chad, and Russia, mainly from grassland, forest, and water, and the area of these main types converted to wetlands was 225,600 km$^2$, accounting for 83% of the total area converted to wetlands in these three countries. Specifically, the total area converted to wetlands in Australia was 111,800 km$^2$, mainly from grassland, bare land, and water, accounting for 82% of the total area of wetlands converted from other land types. The total area converted to wetlands in Chad was 58,400 km$^2$, mainly from grassland, forest, and shrubs, and the area of these three types of conversions accounted for 95% of the total area converted to wetlands in Chad. The total area converted to wetlands in Russia was 100,800 km$^2$, mainly from the forest, grassland, and water, accounting for 91% of the total area converted from other land types in Russia. From 2000 to 2020, the three countries with the largest decreases in wetland areas were Canada, Mali, and Finland, mainly grassland, water, and forest; the area of these three types converted from wetlands was 96,200 km$^2$, accounting for 86% of the total area converted from wetlands in these three countries. Specifically, the total area converted from wetlands in Canada was 176,400 km$^2$, mainly tundra and forest, and the area of these two types of conversions accounted for 83% of the total area of wetlands converted to other land types in Canada. The total area converted from wetlands in Mali was 14,300 km$^2$, mainly grassland, accounting for 95% of the total area of wetlands converted to other land types in this country. The total area converted from wetlands in Finland was 6,900 km$^2$, mainly forest and tundra, accounting for 78% of the total area of wetlands converted to others in Finland.

In this study, the equalities of area changes in two time periods of 2000–2010 and 2010–2020 were evaluated by Lorenz curves to represent the change information at a finer temporal scale. Figure 12a,b represent the Lorenz curves for the increase and decrease in wetland area from 2000 to 2010, and Figure 13a,b represent the Lorenz curves for the increase and decrease in wetland area from 2010 to 2020.

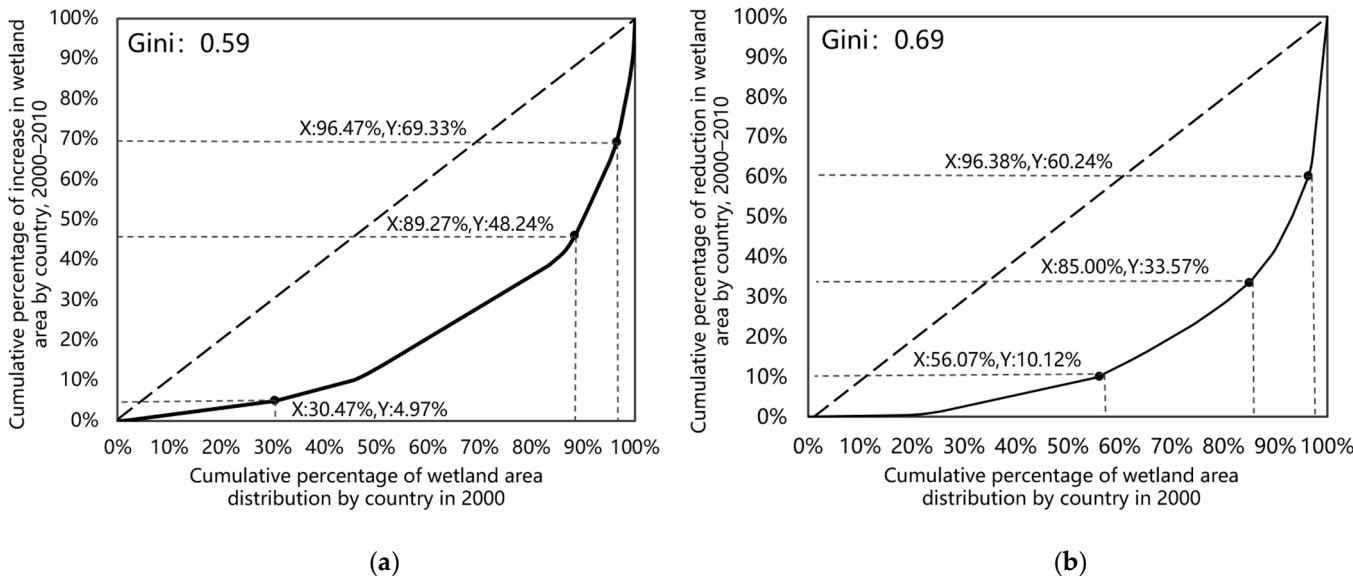

**Figure 12.** Lorenz curves representing the equality of wetland area changes from 2000 to 2010. (**a**). Lorenz curve for the area increase. (**b**). Lorenz curve for the area decrease.

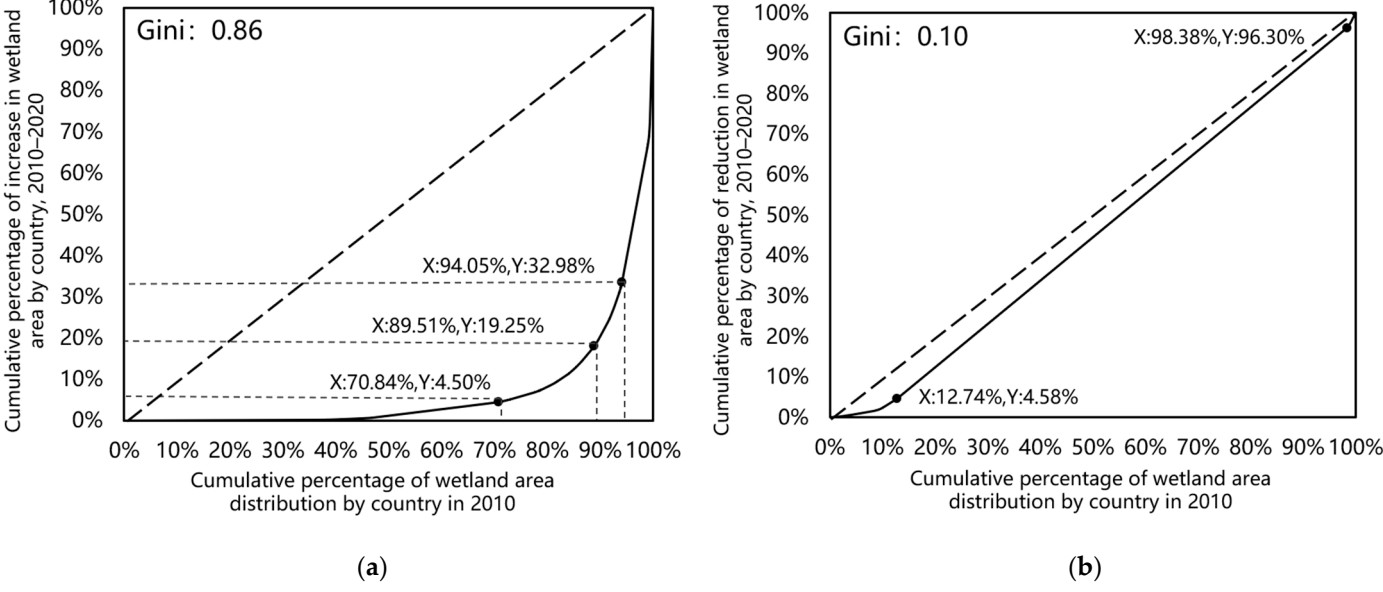

**Figure 13.** Lorenz curves representing the equality of wetland area changes from 2010 to 2020. (**a**). Lorenz curve for the area increase. (**b**). Lorenz curve for the area decrease.

- As shown in Figure 12a, the Lorenz curve has a high curvature, and approximately 10% of countries have more than 50% of the increase in global wetland area. The Gini coefficient is 0.59 (>0.5 indicates a significant inequality), indicating that the increase in global wetland area was uneven. The three countries with the highest increases in wetland area were Russia, Australia, and Bolivia, accounting for 46% of the global increase in wetland area.
- As shown in Figure 12b, the curvature of the Lorenz curve of the wetland area decreased from 2000 to 2010, and approximately 15% of the countries experienced nearly 70% of the decrease in global wetland area. Its Gini coefficient is 0.69 (>0.5 indicates a significant inequality), indicating that the decrease in global wetland area was more uneven. The three countries with the highest decreases in wetland area

were Mali, the Democratic Republic of Congo, and Chad, with a 49% decrease in the global wetland area.

- As shown in Figure 13a, the Lorenz curve has the highest curvature, and approximately 10% of countries have more than 80% of the increase in global wetland area. Its Gini coefficients is 0.86 (>0.5 indicates a significant inequality), indicating that the increase in global wetland area was extremely uneven. The three countries with the highest increases in global wetland area were Australia, Chad, and Kazakhstan, accounting for 60% of the total increase in global wetland area.
- As shown in Figure 13b, the curvature of the Lorenz curve of wetland area decreases for 2010 to 2020 is the smallest, and the curve is close to the perfect equality line because its Gini coefficient is 0.10 (<0.2 indicates perfect equality). The three countries with the largest decreases in wetland area in this period were Canada, Norway, and Finland, with a 97% decrease in global wetland area.

## 4. Conclusions

GL30 data were used in this study to calculate the area, change rate, and importance index of the change in wetlands, as well as the conversion matrix and Lorenz curve at global, continental, climate zone, and national scales in 2000, 2010, and 2020. Using the extracted information on spatial and temporal wetland changes and land type conversion, the characteristics of spatial and temporal changes in global wetlands and conversions between wetlands and other land cover types from 2000 to 2020 were analyzed.

The global wetlands are mainly distributed in the north middle- and high-latitudes and the equatorial middle- and low-latitudes, similar to the analysis results of global-wetland-related datasets and some simulation work [11,29,46]. However, some related studies found that Asia was the continent with the largest wetland area [39], but it was the second largest in the analysis of this study. This may be due to the large areas of seasonally inundated Asian rice paddies [18,39], which may be classified as cropland when producing GL30 data. The following are the main findings of the global wetland changes analysis.

From 2000 to 2020, the wetland area continued to increase, and the net change in global wetland area was +126,400 km$^2$, with a change rate of 4.09%. From 2010 to 2020, the extent of change in wetlands increased significantly, with a net increase of +102,400 km$^2$, the fastest wetland growth period in the last 20 years. Oceania had the largest net increase in wetland area, and its wetland growth was mainly in the central plains of this continent. North America was the continent with the largest decrease in wetland area, and its wetland area decreased mainly in the region near the Gulf Coast of the southern United States, and the region of Canada near Hudson Bay.

From 2000 to 2020, the main land cover types of the conversions with wetlands were forest, grass, water, and tundra, accounting for more than 70% and 80% of the increase and decrease in total wetland area, respectively. The conversion of wetlands to forest, grass, and water occurred primarily in the middle and low latitudes near the equator, especially in the tropical climate zone in South America, and the conversion of wetlands to tundra occurred primarily in the middle and high latitudes of the Northern Hemisphere, especially in the continental subarctic climate zone near Hudson Bay, Canada in North America. In addition, there have been some conversions of wetlands due to human activities, including the conversion of wetlands into cropland (~4600 km$^2$) and artificial land (~3400 km$^2$). According to some relative analysis, it may be affected by urban development, agriculture, and aquaculture land occupation [28,46].

An analysis of the global wetland change equality on a national scale using the Lorenz curve and Gini coefficient showed an inequality in the increase in wetland area from 2000 to 2020, while the decrease in wetland area from 2000 to 2020 was nearly equal. The three countries with the largest increases in wetland area were Australia, Chad, and Russia, which accounted for approximately 60% of the total increase in the global wetland area, and the increase in wetlands mainly from grass, forest, and water accounted for 83% of the total area converted to wetlands in these three countries. The three countries with the largest

decrease in wetland areas were Canada, Mali, and Finland, which had approximately 97% of the decrease in the global wetland area. The decrease in wetlands was mainly converted to grass, water, and forest, accounting for 86% of the total area converted from wetlands in these three countries. Significant changes in wetlands in some countries have also been reported and analyzed in several countries, for example, Canada [28] and Russia [43].

In comparison to previous research, this study comprehensively characterized the distributions and changes of global wetlands, as well as the conversions between wetlands and other land cover types at different temporal and spatial scales, which will help us better understand how wetlands have changed globally and may provide basic information and knowledge for wetland-related research. Further research will focus on analyzing regions with significant changes in wetlands at a higher temporal resolution, such as the Hudson Bay coast of Canada in North America and the Gulf Coast of southern United States, using a combination of remote sensing image data.

**Author Contributions:** Conceptualization, P.T.; methodology, P.T. and M.L.; software, M.L.; validation, P.T., M.L. and X.Z.; formal analysis, M.L.; investigation, M.L.; resources, M.L., X.Z., T.X. and Y.M.; data curation, M.L., X.Z., T.X. and Y.M.; writing—original draft preparation, P.T. and M.L.; writing—review and editing, P.T., M.L, X.Z. and Z.L.; visualization, M.L.; supervision, P.T. and Z.L.; project administration, P.T.; funding acquisition, P.T. and Z.L. All authors have read and agreed to the published version of the manuscript.

**Funding:** This research was funded by the Science and Technology Fundamental Resources Investigation Program of China (2019FY202504), the National Natural Science Foundation of China (41871365; 41930104), and the Sichuan Science and Technology Project (2020JDTD0003).

**Data Availability Statement:** Not applicable.

**Acknowledgments:** We thank the National Geomatics Center of China for providing GL30 data.

**Conflicts of Interest:** The authors declare no conflict of interest.

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
