# Peer review of "Analysis of Spatial and Temporal Variability of Global Wetlands during the Last 20 Years Using GlobeLand30 Data"

_remotesensing, doi:10.3390/rs14215553_

Round 1

Reviewer 1 Report

Based on GlobeLand30 global land cover products (2000, 2010 and 2020), this manuscript analyzed the changes of wetland and their spatial distribution on a global scale in detail. In terms of analysis methods, the authors mainly used the traditional land cover change analysis indexes, and combined the Lorenz curve and Gini coefficient to conduct the balance analysis of wetland change at the national scale. The results of the analysis will help to better understand the global changes of wetlands and provide basic information for wetland related research and conservation. There are no major problems in the data, methods and conclusions of this paper, and it is suggested that it be published after minor modifications. 

A few minor questions are as following:

1. It is GlobeLand30, not GlobalLand30. Please check its spelling in the full text.

2. In the discussion, it is appropriate to discuss the differences between the definition of wetland type in GlobeLand30 and the internationally widely adopted definition of wetland.

3. Refereces [1], [5], [21]-[24], and more. The surnames of some Chinese authors are wrong.

Reviewer 2 Report

The paper identified global distribution and changes of wetlands during the last 20 years and conversion with other land cover types. The results can provide important information for ecologists and climatologists. The following issues should be further considered to improve the quality of the manuscript.

(i)               Page 2: the authors state that “The analysis of changes in the world’s remaining wetlands on a global and regional extent is challenging.” The challenge should be clarified to make it concrete and also how your method tackles it.

(ii)             In Section 2.1, it is necessary to describe the definition differences of wetland type between GL30 and other research work. 

(iii)            The changes represented in Figure 3 are not clear enough. I suggest that each continent should be a subfigure to improve the clarity.

(iv)            In Section 3.2, the comparison between Figure 9(a) and (b) may be needed for the analysis of the changes and conversion from 2010-2020. There are two Figure 8(a) and I guess the second Figure 8(a) should be Figure 8(b).

Reviewer 3 Report

The paper conducted a system analysis on the changes of global wetland during the last 20 years and the changes are estimated using GL 30 data with higher overall accuracies. This topic on wetland changes has been previously studied in isolation by some authors. It is my understanding that the contribution of this paper is to provide more comprehensive information on the global changes of wetlands, including spatial distribution, area change as well as the conversion with other land cover types, and conduct systematic analysis at different spatial scales. I feel the work presented by the authors is very interesting and has clear potential. As I have read the paper I had some questions/suggestions as I summarize below. 

Specific comments:

It is necessary to describe the data processing on the extraction techniques of the change information from the raster data GL30 in Section 2.

In equation (2), some definitions on some variants (S, i) are not given.

In the first paragraph in Section 3.1, more details on the similarity between the distribution characteristics described in this paper and other related work [3,11,39] should be described. 

In Figure 3, the figures of the regions with larger changes should be connected to their location on global map, e.g., by arrow lines. The sub-figure capital should be Figure 3(*), instead of *.

It is better to give the location information of regions by the ranges of longitude and latitude, instead only longitude, e.g., in Asia and South America (15° S–10° N) and in North America (50° N–65° N) in Section 3.2. 

Generally, the text reads rather well. However, there are some small mistakes, awkward phrases, and odd use of words. A language editing is needed.
